# Landslide Susceptibility Mapping of Karakorum Highway Combined with the Application of SBAS-InSAR Technology

**DOI:** 10.3390/s19122685

**Published:** 2019-06-14

**Authors:** Fumeng Zhao, Xingmin Meng, Yi Zhang, Guan Chen, Xiaojun Su, Dongxia Yue

**Affiliations:** 1Key Laboratory of West China’s Environmental System (Ministry of Education), College of Earth and Environmental Sciences, Lanzhou University, Lanzhou 730000, China; zhaofm13@lzu.edu.cn (F.Z.); zhangyigeo@lzu.edu.cn (Y.Z.); gchen@lzu.edu.cn (G.C.); suxj17@lzu.edu.cn (X.S.); dxyue@lzu.edu.cn (D.Y.); 2Gansu Environmental Geology and Geohazards Engineering Research Centre, Lanzhou University, Lanzhou 730000, China

**Keywords:** KKH, landslide susceptibility mapping, logistic regression, random forest, SBAS-InSAR

## Abstract

Geological conditions along the Karakorum Highway (KKH) promote the occurrence of frequent natural disasters, which pose a serious threat to its normal operation. Landslide susceptibility mapping (LSM) provides a basis for analyzing and evaluating the degree of landslide susceptibility of an area. However, there has been limited analysis of actual landslide activity processes in real-time. The SBAS-InSAR (Small Baseline Subsets-Interferometric Synthetic Aperture Radar) method can fully consider the current landslide susceptibility situation and, thus, it can be used to optimize the results of LSM. In this study, we compared the results of LSM using logistic regression and Random Forest models along the KKH. Both approaches produced a classification in terms of very low, low, moderate, high, and very high landslide susceptibility. The evaluation results of the two models revealed a high susceptibility of land sliding in the Gaizi Valley and the Tashkurgan Valley. The Receiver Operating Characteristic (ROC) curve and historical landslide verification points were used to compare the evaluation accuracy of the two models. The Area under Curve (AUC) value of the Random Forest model was 0.981, and 98.79% of the historical landslide points in the verification points fell within the range of high and very high landslide susceptibility degrees. The Random Forest evaluation results were found to be superior to those of the logistic regression and they were combined with the SBAS-InSAR results to conduct a new LSM. The results showed an increase in the landslide susceptibility degree for 2808 cells. We conclude that this optimized landslide susceptibility mapping can provide valuable decision support for disaster prevention and it also provides theoretical guidance for the maintenance and normal operation of KKH.

## 1. Introduction

Landslides are the results of changes in environmental parameters. Under the influence of extreme climatic events, an increasing number of landslides are occurring worldwide, which results in major economic and human losses [1]. The Karakorum Highway (KKH) is an important part of the Asian Highway Network and is China’s gateway to Pakistan and the South Asian subcontinent, as well as being a geopolitical link between China and Pakistan [2]. However, the elevated mountain topography, abundant loose debris, and abrupt heavy rainfall events are leading to severe and frequent geological disasters, such as rock collapses, glacier debris flows, landslides, debris creep, soil creep, and, in rare cases, avalanches [3,4]. Since 1987, over 115 rock avalanches have been generated by very large rockslides or rock falls [5]. In 2005, there was a major earthquake in northern Pakistan, which caused sustained and varying levels of damage to the KKH and induces many landslides [6]. In 2010, a massive landslide in Attabad blocked the river and created a barrier lake more than 20-km long, which flooded the highway and cut off road traffic [7]. By 2011, 150 glacial debris flows had been investigated along the KKH, which caused damaged bridges [4]. All of these disasters result in enormous damage to roads and the surrounding environment. Therefore, it is particularly important to evaluate the geological disaster susceptibility of KKH. But due to the large extent and complex geological conditions in this area, it is difficult to deepen field investigation and assessment work, which means the current disaster assessment of the area only stays in the investigation and evaluation of a single debris flow ditch or a single landslide, such as the investigation of surge-type glaciers [8] and the activity degree evaluation of glacial debris flow [3]. Additionally, due to the lack of historical disaster data, it is difficult to conduct the disaster assessment on a regional scale. Therefore, this research makes up for the lack of historical disaster data and the evaluation gap on the regional scale, using SBAS-InSAR results to optimize landslide susceptibility degrees, which can greatly improve the accuracy of landslide susceptibility assessment at a regional scale. Landslide susceptibility mapping (LSM) provides a basis for geological risk assessment. It is clearly of great importance to conduct regional LSM for the purposes of landslide prevention and mitigation, and landslide management. The need for quantitative risk management is increasing, and, therefore, quantitative methods are needed to evaluate landslide hazards and risk zoning. After the emergence of GIS (Geographic Information System) technology, numerous quantitative approaches to landslide susceptibility were developed, such as Logistic Regression, Analytic Hierarchy Process, Artificial Neural Networks, Support Vector Machines, and Random Forest. All of these methods have been shown to have specific advantages in different study areas and for different sets of factors involved in disasters [9,10,11,12,13]. However, the accuracy of LSM is not just related to the method itself, but also relates to the input training dataset including the historical landslides and landslide predisposing factors. In order to compare the accuracy of different evaluation methods, we chose logistic regression and random forest models to evaluate landslide susceptibility of KKH separately. In order to reduce the misclassification of LSM caused by the incompleteness of historical landslide data, we optimize LSM by using the InSAR results closely related to landslide occurrence to improve the reliability of LSM.

Interferometric Synthetic Aperture Radar (InSAR) technology has been widely used in mapping high accuracy slope movements, which is of great relevance for the landslide risk assessment [14,15]. The use of space-borne radar measures has proven to be a reliable approach for improving landslide monitoring and identification for the landslide risk assessment. Oliveira used PS-InSAR to assess its capacity in landslide susceptibility on a regional scale [16]. Piacentini also verified the accuracy and superiority of PS-InSAR in landslide susceptibility assessment [17]. In particular, the Small Baseline Subsets InSAR (SBAS-InSAR) technology overcomes the limitations of time incoherence and avoids the long temporal separation, spatial incoherence, and atmospheric effects in traditional interferometry methods, which produces land deformation results that are more continuous in time and space [18,19]. The method is suitable for monitoring both slow linear and nonlinear deformation of long sequences and is widely used in the monitoring of land subsidence, earthquakes, active faults, and glacier migration, as well as of slope instability such as creep and landslides [20,21,22,23,24]. 

In this study, logistic regression and Random Forest models were used to evaluate the application of LSM to the KKH, with the aim of comparing the accuracy of the mathematical model and the machine learning model. The accuracy of the results was evaluated by verifying the correct classification rate of the sample set and the Area Under Curve (AUC) value of the ROC (Receiver Operating Characteristic) curve. However, there was still potential for the misclassification of the results of the LSM. For example, if a landslide occurs in a certain area, the susceptibility map may still indicate that the area is less susceptible. To reduce the probability of misclassification, we used the 2016–2017 SBAS-InSAR deformation results to correct the errors, which allows the optimized susceptibility map to provide more reliable decisions for land use and landslide prevention and mitigation.

## 2. Study Area

This study area is identified by the 10-km buffer of the Karakoram Highway. It runs from Shufu County in Kashgar to the Khunjerab Pass in China (Figure 1). It is located in the western part of the Himalayan orogenic belt caused by the India-Asia collision [25]. A major component of the environment of the highway is the occurrence of mountain glaciers, which result in major glacial debris flows and rock avalanches [3]. Topographically, the KKH is characterized by deep valleys and rugged mountains [6], with elevations ranging from 1280 to 6961 m. Strong tectonic movements and rapid uplift have produced deep, narrow river valleys that provide dynamic conditions, which promote the occurrence of geological disasters [26]. Therefore, geological disasters such as mudslides, landslides, and flash floods occur frequently along the KKH, causing serious damage to the KKH. 

Quaternary loose deposits are the main surface strata along the KKH. The exposed sedimentary rocks and metamorphic rocks are mainly Proterozoic, Paleozoic (Carboniferous and Permian), and Mesozoic (Triassic and Cretaceous). There are sporadic outcrops in every section within areas of magmatic rocks, which are mainly Caledonian granite, Himalayan granite, and diorite. The overall high altitude environment results in a cold, dry, and drought-prone climate. Annual precipitation is less than 200 mm including 70% of which falls during July to September [26]. Consequently, the region spanned by the KKH has very low vegetation coverage.

## 3. Data and Methods 

### 3.1. Data and Variables 

Using remote sensing image interpretation, field surveys, and the collection of historical data, a total of 44 landslides along the KKH were identified. These historical landslides are mostly distributed along both sides of the highway, with an area ranging from 3 × 10^−6^ to 2.4 × 10^−4^ km^2^ (Figure 1). The mountains on both sides are steep, and most of the mountain tops have perennial snow, which makes the mountains prone to landslides under the action of glacier melt water in the summer. Of these historical landslides, they are divided into five types, including 23 complex slides (52.27%), 10 rock falls or topple (22.73%), seven superficial instability events (15.91%), and four creep (9.09%). Using GIS, the landslide area was converted into 4546 landslides points, and an equal number of non-landslide points were randomly generated from the area. Landslide points include training datasets to train models and validation datasets to validate models. In this study, 80% of the total landslide points, comprising 3637 landslide points, were selected randomly for model training, and the remaining 20% of points, comprising 909 points, were used to validate the accuracy of the training model.

There is no general criterion for selecting the independent variables for landslide susceptibility mapping. The principles are that the variables must be operational, non-uniform, measurable, and non-redundant [9]. In this study, the original data included the following: a geological map from the China Geological Survey (scale 1:500,000), a 30-m SRTM DEM, 30-m Landsat 8 images, and 0.25° 3B43 TRMM images. The variables considered for the LSM of the KKH were: elevation, slope, aspect, profile curvature, planar curvature, and Topographic Wetness Index (TWI) derived from the DEM. Normalized Difference Vegetation Index (NDVI) derived from Landsat 8 images, annual precipitation derived from TRMM images, and lithology and fault data derived from the geological map. 

Since there may be a high correlation between the independent variables used in the linear regression model, the model estimation may be distorted or be difficult to estimate accurately. This is because the data may not contain sufficient information to describe the problem under consideration when the factors are highly correlated with each other. Therefore, before model operation, a multicollinearity analysis of the independent variables is performed, and the variables with significant collinearity are eliminated. Commonly-used collinearity diagnostic indicators include the variance inflation factor (VIF) and tolerance (TOL) [27], which are reciprocal. In general, VIF > 10 (TOL < 0.1) indicates that the selected variable has multiple collinearity. Multicollinearity diagnostics of the factor data influencing landslide occurrence were obtained using SPSS (Statistical Product and Service Solutions) software. In this study, the TOL values of the above 10 variables are all <1, as shown in Table 1, which ensures the independence of the factors and enables them to participate in model estimation. The 10 variables were classified using the natural break point method, displayed hierarchically, and assigned to the training datasets to facilitate the operation of the model (Figure 2).

The choice of mapping unit is critical for LSM. In this study, a 100*100 m moving window was used to calculate mean value for each numerical variable [28]. For these variables, the average value inside a 100 × 100 m cell was used in the LSM.

### 3.2. Landslide Susceptibility Models

#### 3.2.1. Logistic Regression Model

The logistic regression model is a generalized linear regression method. It uses dichotomous variables such as 0 and 1 determined by one or more independent variables to analyze the problem [9]. Logistic regression has been widely used in landslide space prediction research because it uses simple forms to address complex nonlinear problems, and can generate unbiased results. The approach uses the maximum likelihood method to construct the relationship between the predictor and the two-category result, in order to ensure that each point is the best fit. In geological hazard susceptibility analysis, it is often used to describe the relationship between binary dependent variables (usually 0/1 for the occurrence/non-occurrence of a geological disaster) and several independent variables. The independent variable can be continuous or discrete and it does not have to be normally distributed. The model is expressed below.

(1)$$P=\frac{1}{1+\exp\left(-\left(\beta_{0}+\beta_{1}X_{1}+\beta_{2}X_{2}+\cdots +\beta_{n}X_{n}\right)\right)}$$

In this case, P is the probability of landslide occurrence. $\beta_{i}$ is the logistic regression coefficients based on training samples, i=0, 1, 2, ⋯, n, and $X_{i}$ is the independent variable, i=0, 1, 2, ⋯, n [29].

In this study, training landslide points and an equal number of random non-landslide points were selected as statistical samples for evaluating landslide susceptibility in the KKH. The attribute values of the independent variables were assigned to each sample point, and all of the sample values were imported into SPSS to perform binary logistic regression analysis. 

#### 3.2.2. Random Forest Model

The Random Forest model, which originates from the bagging algorithm of Breiman [30], is an integrated learning mode based on random sampling. Essentially, it is the product of integrating many decision trees into forests and combining them to predict the end result. Model building is a process of decision tree combination. The decision tree is similar to the flowchart of the tree structure, which is a recursive process from top to bottom. Starting from the nodes of the tree, the optimal features are selected for different internal nodes, and the corresponding branches are determined according to the test output. The final results are derived from the nodes of the decision tree leaves.

The Random Forest algorithm can process datasets with large dimensions and large data volume and it has a high generalization ability. Compared with other statistical learning methods, it is not susceptible to overfitting, and the prediction accuracy is improved based on the criterion of no further significant improvement in the calculation. Moreover, the results are robust in the case of missing and unbalanced data [31]. It has been widely used in data classification and management, and has performed well in LSM [13]. In this study, Random Forest modules in the R language environment were used to evaluate the landslide susceptibility.

#### 3.2.3. Verification of Model Accuracy

The following steps are followed in order to verify the accuracy of the model. First, a rationality test is carried out based on the distribution of the actual occurrence of the landslide points in each landslide susceptibility level. However, subjectivity is involved in selecting landslide points for testing. In order to maximize the stability of the established model, the assessment points consist of 20% of the verification points that are not involved in model calculation. The test for the rationality of the model is based on the criterion that the percentage of the group of landslide points falling within a landslide-prone area is the largest.

The second step is to use the Receiver Operating Characteristic (ROC) curve to test the accuracy of the model evaluation, which is an effective graphical method for evaluating the validity of classification. The ROC curve is based on a series of different two-category methods, with the true positive rate (sensitivity) as the Y-axis and the false positive rate (1-specificity) as the X-axis. In this case, the true positive rate is the proportion of landslide units correctly predicted, and the false positive rate is the proportion of the units that were not correctly predicted as landslide units [32]. The closer the ROC curve to the upper left corner of the graph, the higher the accuracy of the test is. The area under the ROC curve is the Area under Curve (AUC) value, which is an indicator of the accuracy of the model. The range of values of AUC is [0.5, 1]. The larger the value, the better the effect of model discrimination.

### 3.3. SBAS-InSAR

InSAR technology has been widely used in the early identification of landslides due to its advantages of independence of weather conditions as well as its wide range of monitoring and high-precision monitoring. The SBAS-InSAR method determines surface deformation over time by processing the interferogram with a shorter baseline and smaller decorrelations, which makes the deformation results denser and more reliable [21].

In this study, a set of 28 sentinel-1A images from 2016 to 2017 were processed in the SARScape module. The first step was to generate the connection graph. In order to generate the SAR image pairs, the maximum normal baseline and temporal baseline were set to 110 m and 90 days, respectively (Figure 3), and 76 interferogram pairs were generated. The most critical step is the interferogram process, conducted using a Goldstein filter, which can improve both the measurement accuracy and phase unwrapping in order to maximize the signal/noise ratio of the interferograms [33]. The Minimum Cost Flow method (MCF) [34], with a coherence threshold of 0.35, was used for unwrapping. Thus, some interferogram pairs with low coherence and unwrapped phase error were removed from the connection graph. The next step was to use the Ground Control Points (GCPs) to execute the orbital refinement and phase re-flattening process for the remaining pairs. The selection criterion for GCPs is that they are located far from the area of deformation and in better unwrapped phase regions. In addition, in order to ensure the accuracy of the GCPs, the coherence value cannot be <0.7 [35,36]. A cubic inversion model was used to remove the residual topography and derive the preliminary displacement. Subsequently, the Atmosphere Low Pass Size was set to 1000 m and the Atmosphere High Pass Size to 365 days in order to remove the atmosphere residual error. Lastly, the least squares solution obtained by the Singular Value Decomposition (SVD) method was used to estimate the nonlinear deformation of the time series.

The SBAS method obtains surface deformation values along the line of sight (LOS). For mountainous areas, the rate of deformation in the LOS direction is insufficient for reflecting the true deformation of the slope [20]. The deformation rate of the LOS direction is converted to the deformation rate along the slope, using the following formulas.
(2)$$V_{slope}=\frac{V_{los}}{Index}$$
(3)$$\mathrm{Index}=n_{los}\times n_{slope}$$
(4)$$n_{los}=\left(-\sin\theta \cos\alpha_{s},\;\;\sin\theta \sin\alpha_{s},\cos\theta \right)$$
(5)$$n_{slope}=\left(-\sin\alpha \cos\varphi ,-\cos\alpha \cos\varphi ,\sin\varphi \right)$$

In this case, V_slope_ is the deformation rate along the slope angle, V_los_ is the deformation rate of the LOS direction, α is the aspect, ψ is the slope, θ is the radar incident angle, α_s_ is the angle between the direction of the satellite orbit and true north (which is the radar satellite flight direction), ascending data are negative, and descending data are positive [37].

### 3.4. Refinement

The purpose of refinement is to increase the accuracy of LSM, and to minimize the results of misclassification in order to obtain more accurate assessment results for landslide-prone areas. In order to achieve this goal, we used the SBAS-InSAR slope deformation results to integrate the LSM by building the contingency matrix between the landslide susceptibility degree and the SBAS deformation rate (Table 2). The velocity intervals were determined based on the standard deviation (δ = 15 mm/yr) of the V_slope_ [28]. The velocity intervals were determined to increase the susceptibility degree from level 1 to 5. The greater the value of V_slope_ is, the higher the landslide susceptibility degree is. The susceptibility degree cannot exceed 5. This integration reduces the probability of misclassification, and it results in areas prone to landslides, which have a low susceptibility assessment degree being assigned a high susceptibility degree.

## 4. Results

### 4.1. Logistic Regression Results

The binary logistic regression model, using multiple iterations, was implemented in SPSS to obtain the landslide susceptibility assessment values (Table 3). B is the regression coefficient of each variable in the model. S.E. is the standard deviation. Wald is the Chi-Square statistic. Sig. is the significance level. Notably, there is a statistically significant difference between the regression coefficients of the variables. The significance values of most of the variables are less than 0.1, which indicates that the logistic regression results pass the significance test. The coefficient of each variable can be used as a measure of the relative importance of the independent variable, with a positive coefficient indicating that the independent variable is positively related to landslide susceptibility [27]. In other words, the larger the value of the factor, the greater the degree of proneness to land sliding. A negative coefficient indicates that the independent variable has an inhibitory effect on the landslide susceptibility. For example, the greater the vegetation cover, the less likely a landslide is to occur. Reference to Table 3 shows that the slope has a greater effect on the development of landslide proneness than any other variable, which indicates that slope failure is directly associated with topographic characteristics. Slope is positively correlated with landslide occurrence since it is related to shear stresses acting on the displacement of the hill slope. When the soil layer is sufficiently thick, the greater the slope is, the more unstable the slope will be [38].

The natural breaks method was used in the GIS environment to classify the landslide susceptibility probability. The classification is based on natural groupings inherent in the data, and the classification interval is identified, which provides an optimum grouping of similar values and maximizes the differences between classes [27]. We used this approach to divide the data into five categories: 0~0.13, 0.13~0.30, 0.30~0.51, 0.51~0.73, and 0.73~1, which respectively represent very low, low, moderate, high, and very high landslide susceptibility (Figure 4). The performance of the model was evaluated using the ROC curve. The AUC value is 0.793, which satisfies the requirement for landslide susceptibility accuracy. The remaining 909 landslide points were then used to verify the model accuracy. There are 578 points (63.59%) in the high and very high ranges.

### 4.2. Random Forest Results

We then used the Random Forest model implemented in R to obtain landslide susceptibility evaluation results, which indicated the relative importance of each evaluation factor in the model (Figure 5). The Random Forest method has two indicators for measuring the contribution of variables to landslide susceptibility: Mean Decrease in Accuracy and Mean Decrease in Gini [39]. The larger the value of both indexes is, the greater the importance of the corresponding variable is. Reference to Figure 5 shows that the importance of elevation and slope is relatively high, which is consistent with the results of the logistic regression.

The natural breaks method was again used to classify the landslide occurrence probability into five categories: 0~0.11, 0.11~0.27, 0.27~0.64, 0.64~0.80, and 0.80~1, which corresponds, respectively, to very low, low, moderate, high, and very high (Figure 6). The performance of the model was evaluated using the ROC curve. The AUC value was 0.981, which indicates that the susceptibility result is near-perfect. We used the remaining 20% (909 historical landslide points) to verify the accuracy of the model, and 898 (98.79%) were within the range of high to very high landslide susceptibility.

The AUC value of the evaluation results and the correct classification ratio of the verification points demonstrate that the Random Forest evaluation results are superior to those of the logistic regression. Therefore, we chose the Random Forest evaluation results for optimization of the landslide susceptibility degree.

The results of both the logistic regression and Random Forest models show that the KKH is highly prone to landslides in the areas of the Gaizi Valley and Tashkurgan Valley. In the Gaizi Valley, where landslides occur frequently, the landslide susceptibility degree is mainly high and very high. The geological conditions in the area are complex, faults are developed, and earthquakes are frequent. Under the action of short-duration heavy rainfall events and meltwater from ice and snow, landslides including rock falls, complex slides, superficial instability events, and creep occur frequently, which seriously obstruct the KKH and prevent its normal operation. The other landslide prone area is in the Tashkurgan Valley where the sides are steep and the rock masses are fragmented. The lithology is mainly sericite and turquoise schist, with well-developed joints and fractures, which makes it susceptible to land sliding under the action of water and wind. In addition, in the evaluation results of the Random Forest model, there is an area with a high degree of landslide proneness in the mountainous area on the west side of Tashkurgan County. Notably, a large mudslide occurred there in past, and the entire county is located on a huge debris flow fan. Therefore, engineering work should potentially be undertaken in the area in order to minimize the impact of mountain landslides. From Tashkurgan County to the south, with the gradual increase in altitude, the mountains on both sides of the KKH are affected by ice and snow. The repeated frost heave reduces the stability of the rocks and soils, which makes the area prone to landslides.

### 4.3. SBAS-InSAR Results

The LOS direction deformation rate result was obtained using 0.7 as the coherent threshold (Figure 7a). A total of 6,185,930 coherent targets were obtained, with an average density of 773.82 km^−2^. The LOS direction deformation rate result was −82.46 to 142.48 mm/year and the stability threshold was chosen as −3 to 3 mm/year. The LOS direction deformation result was converted to the along-slope direction deformation result using a transformation formula (Figure 7b). A total of 79,003 slope deformation points were obtained, with a density of 98.82 km^−2^. The maximum slope deformation rate was −473 mm/year, with 0 to −15 mm/year as the threshold stability point. In addition, 94.67% of the slope deformation points are defined as stable. The stability threshold was set as the deformation rate standard deviation. Due to the incoherence caused by regional glaciers and water bodies, the results of SBAS deformation do not cover the entire study area. The area of statistically reliable SBAS deformation results represents 23.16% of the total area, including along the KKH. Field verification of the V_slope_ deformation shows that the geological disasters include 117 landslides, 79 subsidence, and six glacial movements (Figure 7b). The statistic shows that only 22.78% of newly identified landslides by SBAS results fall into the high and very high susceptibility degrees by using a random forest model. Therefore, the results are reliable and can be used to optimize the landslide susceptibility evaluation results. The resulting slope deformation map highlights clusters of deformation points in the following localities: Gaizi Valley, Blumkou Reservoir, Tashkurgan County, and the Khunjerab Pass. However, this distribution is inconsistent with the distribution of historical landslide points, and, therefore, it is necessary to refine the LSM results. Additionally, since the SBAS monitoring results include landslides and subsidence, this study only considers the assessment of landslide susceptibility, the V_slope_, which is defined as removed ground subsidence, and is then optimized for LSM using the V_slope_ after removing the subsidence points. 

### 4.4. Refining the Results

To refine the results, the slope direction deformation rate was converted to the grid values at a 100-m resolution, and the contingency matrix was then used to optimize the landslide susceptibility degree map obtained by the Random Forest model (Figure 8a). Compared with the evaluation results of the Random Forest model, the percentage change of the cells of each degree is not clear in the new susceptibility degree map, and this also illustrates that most of the landslides identified by the SBAS-InSAR method are in the evaluation results. In order to more directly reflect this, the newly generated LSM and the LSM using the Random Forest model were used to compare the degree of difference for each cell (Figure 8b). There are three areas with obvious landslides with increased landslide susceptibility. The percentages of cell degrees in the original and new LSM results are listed in Table 4. Class 1 comprises 56.91% of the cells, class 2 comprises 19.88%, class 3 comprises 10.68%, class 4 comprises 6.77%, and class 5 comprises 5.76%. Calculation of the difference between the new LSM assessment results and the original results revealed that there are 2808 cells with an increase in landslide susceptibility. Among them, 1387 cell susceptibility degrees increased by one, 1084 cell susceptibility degrees increased by two, 305 cell susceptibility degrees increased by three, and 32 cell susceptibility degrees increased by four.

### 4.5. Results of Specific Cases

Three areas with pronounced increase in landslide susceptibility degree (numbered 1–2) are illustrated Figure 8b. We now present a detailed analysis of these three areas. Locality 1, on the west side of Blumkou Reservoir, is a hilly area composed mainly of Quaternary loose deposits. It is substantially affected by the action of wind and rain. Around the reservoir, except for a small part of the shore, the slope is steep, and most of the slope is steeper than 30°. The physical and mechanical parameters of the soil are reduced after long-term immersion softening of the bank slope. As the water level changes reciprocally and the waves are eroded, the soil becomes steeper and the stability decreases. Eventually, a certain scale of collapse, local slip, and other instability events occur. The deformation of SBAS shows that the deformation rate is higher, and assessment of the degree of landslide susceptibility after optimization by the SBAS-InSAR reveals an increase (Figure 9).

Locality 2 is near the Muztag Mountains and glacial moraines are widely distributed. Tectonic activity in the area is strong, earthquakes are frequent, and it is prone to moraine landslides. Previous work indicated that a landslide, in this case, was triggered by an earthquake. As a result of extra-seismic forces, the morainic material in the landslide area began to be mobilized and the structure of the moraines beneath the glaciers in the vicinity of the landslide was affected. Cracks appearance and melting began to affect the stability of the upper layer of the moraines, which results in accelerated movement and damage to the road. The latest SBAS results show that there are multiple deformation centers, which indicate that there is a greater possibility of landslides in areas where landslide susceptibility is low. The optimized LSM shows this difference and corrects the low landslide susceptibility area to high landslide susceptibility area (Figure 10).

## 5. Discussion

This research object is the landslide susceptibility evaluation of KKH. There are few studies on the maximum limit range of damage caused by disasters on both sides of linear highways, so there is no uniform standard for road boundary range, but is only considered for the distribution range of historical landslides and the possible damage to the road. Most of the historical landslides in KKH are within the 10-km buffer zone. Exceeding this range, most of them are glacially covered areas. Because it is far away from the highway, even if a large landslide occurs, it is difficult to cause damage to the highway. Therefore, this study selects the 10-km buffer zone on both sides of KKH as the actual research scope.

When we obtained the LSM by logistic regression and random forest, it is necessary to evaluate its accuracy. The most straightforward way is to calculate the correct classification ratio. To do this, we need to use a testing set to verify the model’s ability to discriminate against new samples, and then use the testing error as an approximation of the generalization error. In general, we assume that the verification sample is also obtained by sampling independently from the true distribution of the sample. Additionally, the verification set should be as mutually exclusive as possible to the training set. In addition, the division of the training set and the verification set should maintain the consistency of the data distribution as much as possible, and avoid the influence of the data division process in the final result. Generally, approximately 2/3 to 4/5 samples are typically used for training and the remaining samples are used for testing.

The optimization of LSM is only effective when the SBAS deformation indicates the landslide occurrence. In fact, the slope deformation recognized by the InSAR method is generally a precursor to landslide occurrence, when the slope deformation rate has an accelerated process in time series analysis. It generally indicates the occurrence of the landslide [20,40]. Through field investigation and verification of InSAR results, we found that the landslide state of activity has been determined through V_slope_. Essentially, the greater the V_slope_ is, the more unstable the slope is, and the more likely the landslide occurs. It is demonstrated that InSAR results can greatly improve the monitoring capacity of very slow landslides. Therefore, the InSAR deformation monitoring results can provide important evidence for early identification and susceptibility evaluation of landslides [16,17].

In the SBAS-InSAR results, the sentinel-1A satellite used in this study has a 12-day revisiting time, which makes the InSAR method suitable only for slow and very slow earth surface movements. Thus, it fails to effectively monitor rapid movements such as rock falls and debris flow. Due to this inherent limitation of the InSAR technology, there is no effective SAR monitoring of vegetated and glacier covered areas. The L-band sensor reduces the temporal incoherence effect caused by vegetation coverage and improves the penetrating ability of the radar signal [28], whereas the X-band sensor is suitable for the monitoring of glaciers [41]. The sentinel-1A C-band, however, combines the advantages of the L-band and X-band sensors to reduce the spatial and temporal decorrelation, and provides an improved monitoring of vegetation and glaciers. 

In this study, logistic regression and the Random Forest model were used to conduct LSM, but the results have several limitations that cause the misclassification. One is related to the quality of data associated with each landslide predisposing factor and the other is the quality of the landslide inventory. Regarding the last one, only 44 landslides mapped due to the harsh environment of KKH exist for this study area, and they cannot completely represent the total number of historical landslides. This resulted in a large misclassification error for the LSM, and demonstrates the necessity of refining the LSM using SBAS results. Notably, the new LSM, combined with the SBAS results, reduced the misclassification in which terrain affected by slope deformation was classified with very low and low susceptibility. Another issue is that the LSM only portrays the predicted landslide distribution in areas and does not depict dynamic deformation processes over time. However, changes in landslide activity over time is a primary concern for decision makers [42]. The new LSM combined with the SBAS results can reveal the actual activity state of landslides and should be used for preliminary landslide mapping and quantitative risk management at a regional scale [43]. 

In summary, LSM is a fundamental tool in landscape management and can be used to evaluate the risks for people and infrastructure areas prone to landslides. Both the LSM and the SBAS deformation results individually can achieve the required goal, but the combination of the two methodologies allows the LSM to be optimized and refined. The LSM optimized by the SBAS results led to an improvement in the susceptibility ranking of a portion of the KKH over the LSM obtained by the Random Forest model. Thus, we suggest that LSM, optimized by the use of SBAS, is potentially very useful for effective land use management and planning activities in the area of the KKH.

## 6. Conclusions

In this study, logistic regression and the Random Forest model were used to evaluate the accuracy of LSM of the KKH. Subsequently, SBAS-InSAR results were used to improve the accuracy of the LSM. The purpose of this combination of techniques is to further improve the accuracy of the evaluation results by taking into account recent landslide events in order to reduce the potential landslide risk. 

LSM was obtained by using logistic regression and random forest methods, respectively, which were divided into five categories: very low, low, moderate, high, and very high. By comparing the correlated classified rate of the validation data set and the AUC value of the ROC curve, the random forest values reached 98.79% and 0.981%, respectively. Therefore, the random forest result was selected to participate in the LSM optimization. Then, we create the contingency matrix based on both the susceptibility degree and the V_slope_ after removing the subsidence rate to optimize LSM. The new LSM showed that there were 6595 cells with the landslide susceptibility degree increased.

Our research has a significant implication for the refinement of landslide susceptibility mapping, especially in areas where the SBAS method is suitable and available. This optimized landslide susceptibility mapping can provide valuable decision support for disaster prevention, mitigation, and management along the KKH.

## Figures and Tables

**Figure 1 sensors-19-02685-f001:**
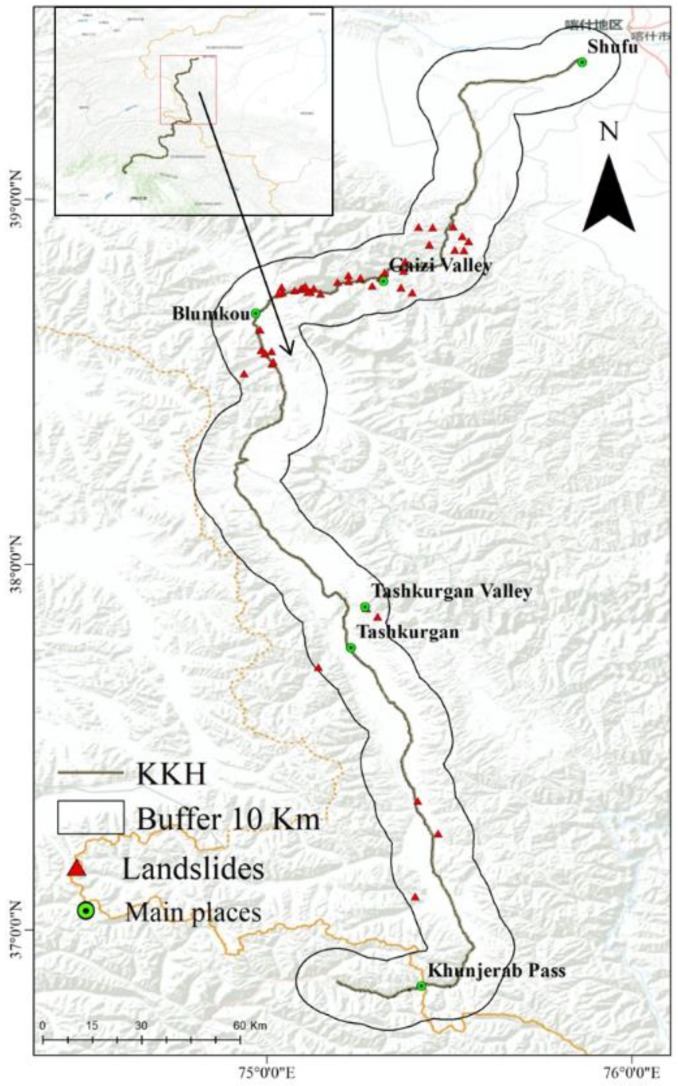
The study area and distribution of landslides (The yellow line is the border between China and Pakistan. The red line on the up-right is the border of Chinese provinces).

**Figure 2 sensors-19-02685-f002:**
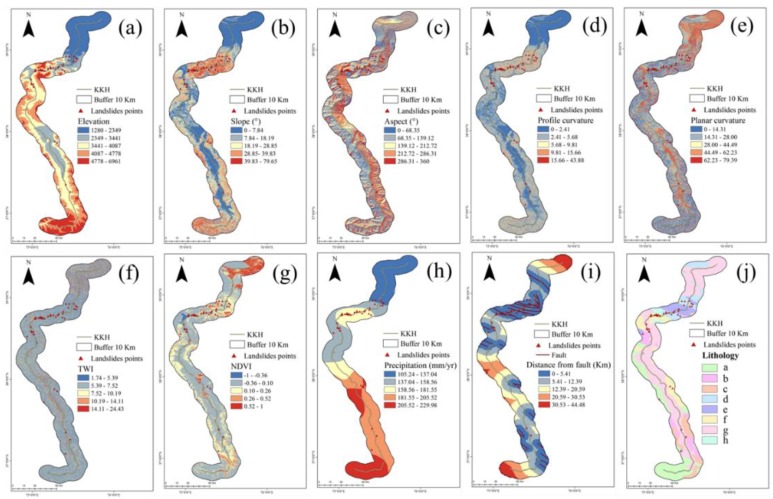
Landslide predisposing factors: (**a**) is the elevation, (**b**) is the slope, (**c**) is the aspect, (**d**) is the profile curvature, (**e**) is the planar curvature, (**f**) is the TWI, (**g**) is the NDVI, (**h**) is the precipitation, (**i**) is the distance from the fault, and (**j**) is the lithology. In (j), a represents plagioclase, monzonite, potassium feldspar granite, diorite, granodiorite, b represents calcareous siltstone, quartz sandstone, phyllite, clastic rock containing pyrite crystals, c is clastic rock sandwiched limestone, occasionally sandwiched siltstone, quartz sandstone, d is gray mudstone, sandstone, black argillaceous siltstone, gray sandy mudstone, limestone, e is gray medium thick, massive limestone, bio-limestone, marl, variegated conglomerate, f is sericite green clay schist, quartz schist marble, quartz marble, dolomite quartz schist, g is gravel, sand, sand soil, gravel soil, sub-clay, and h is snow area.

**Figure 3 sensors-19-02685-f003:**
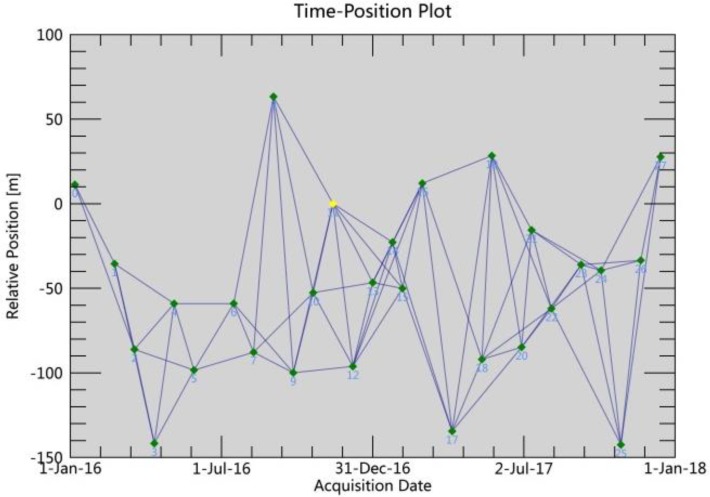
Spatial-temporal baselines of the generated interferograms using SBAS-InSAR (each dot represents an interferogram).

**Figure 4 sensors-19-02685-f004:**
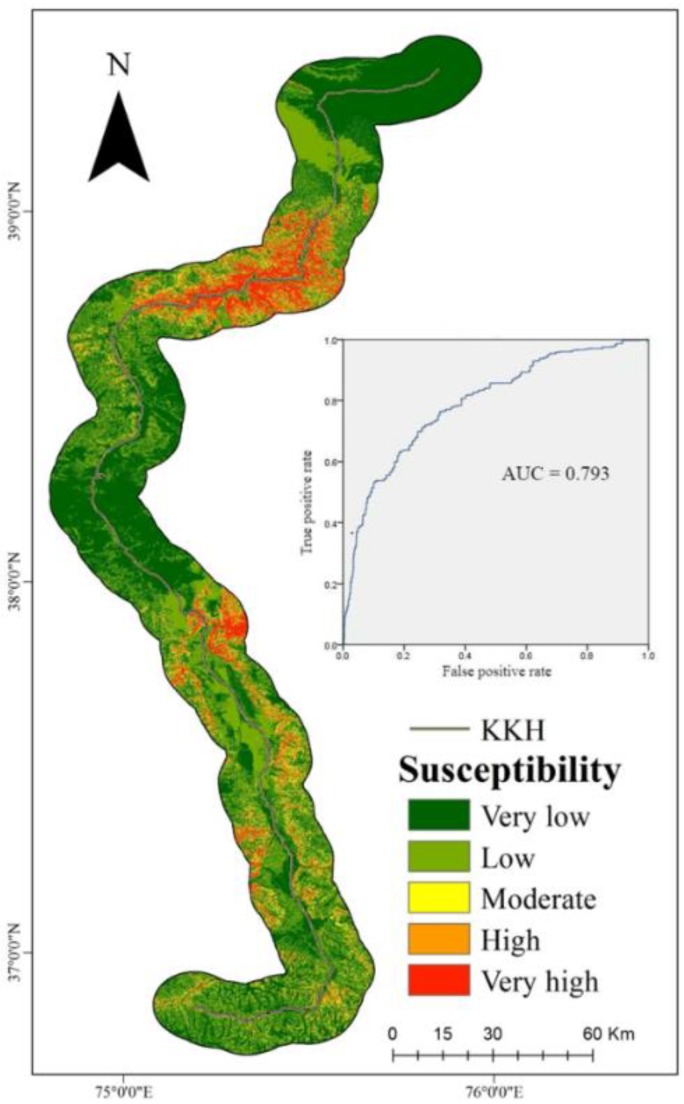
Landslide susceptibility map of the KKH and the related ROC curve, using logistic regression.

**Figure 5 sensors-19-02685-f005:**
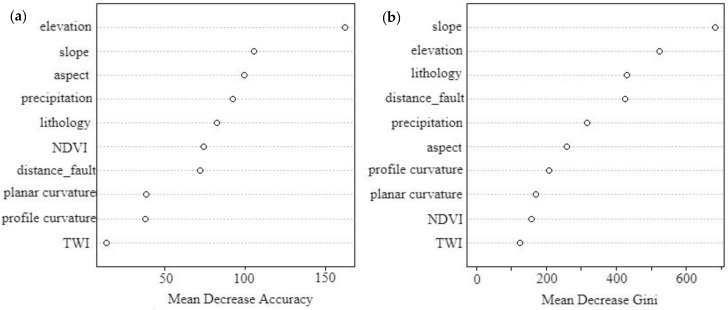
Random forest results: analysis of the relative importance of the variables. (**a**) is Mean Decrease Accuracy, (**b**) is Mean Decrease in Gini.

**Figure 6 sensors-19-02685-f006:**
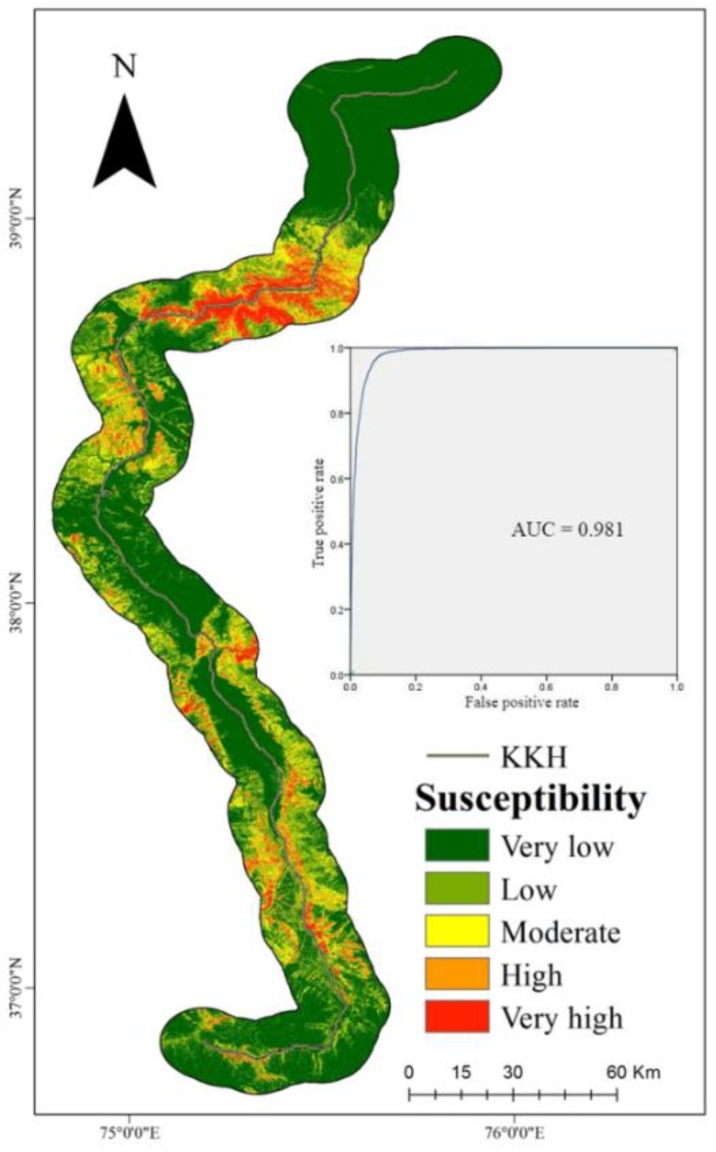
Landslide susceptibility map of KKH and the related ROC curve using the Random Forest model.

**Figure 7 sensors-19-02685-f007:**
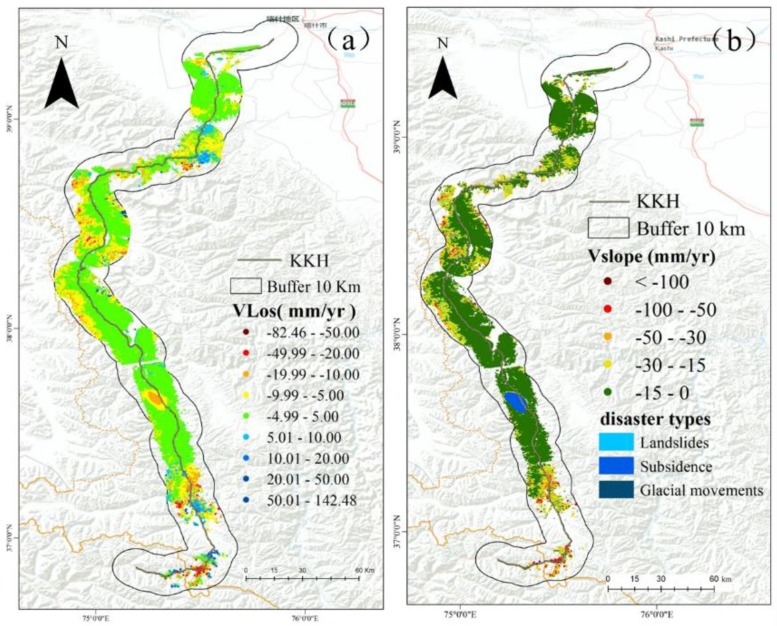
Ground deformation velocity map using SBAS-InSAR along the KKH. (**a**) Deformation velocity along the LOS direction. (**b**) Deformation velocity along the slope direction.

**Figure 8 sensors-19-02685-f008:**
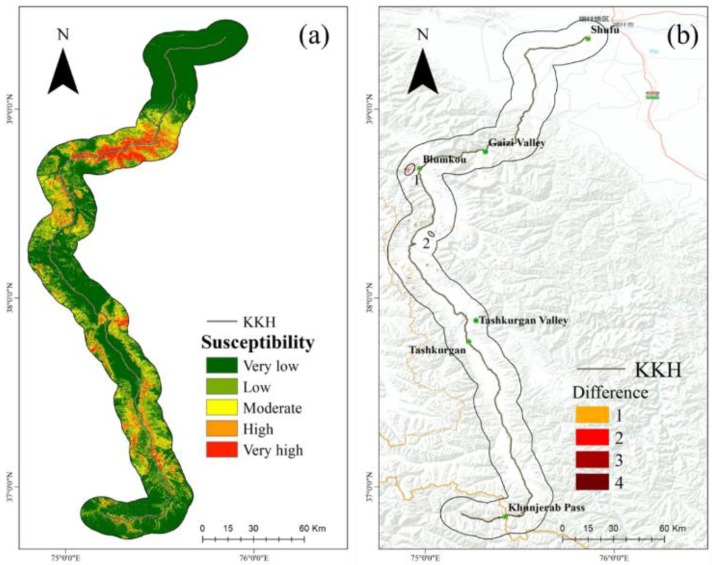
(**a**) New landslide susceptibility mapping results after the application of the correction matrix. (**b**) Difference between the original landslide susceptibility degree and the new landslide susceptibility degree for each cell. The circled areas labelled 1–3 correspond, respectively, to the Blumkou Reservoir, the Muztag Mountains, and an area of mining activity.

**Figure 9 sensors-19-02685-f009:**
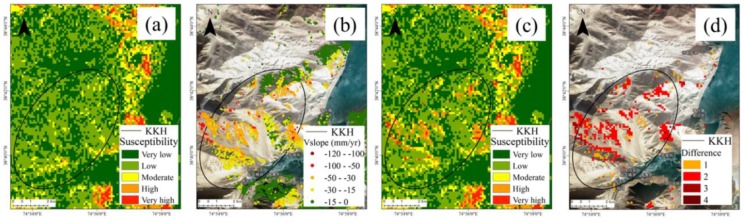
Results of landslide assessment for locality 1 (west of Blumkou Reservoir). (**a**) Landslide susceptibility assessment results obtained using the Random Forest method. (**b**) SBAS deformation rate values along the slope direction. (**c**) Optimized landslide susceptibility assessment results. (**d**) Mapping results of the difference in the degree of landslide susceptibility assessment before and after optimization.

**Figure 10 sensors-19-02685-f010:**
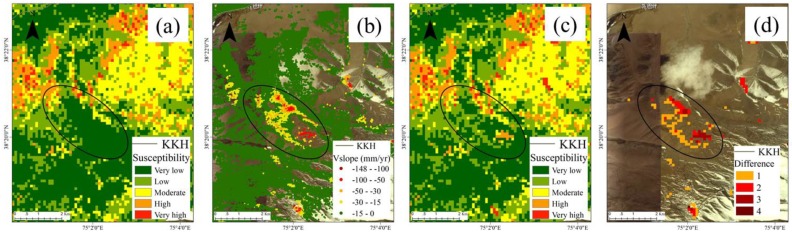
Results of landslide assessment for locality 2 (Muztag Mountains). (**a**) Landslide susceptibility assessment results obtained by the random forest method. (**b**) SBAS deformation rate values along the slope direction. (**c**) Optimized landslide susceptibility assessment results. (**d**) Mapping results of the difference in the degree of landslide susceptibility assessment before and after optimization.

**Table 1 sensors-19-02685-t001:** Multicollinearity diagnostics for the variables used in this study.

Variable	TOL	VIF
Elevation	0.611	1.638
Slope	0.576	1.735
Aspect	0.989	1.011
Profile curvature	0.801	1.248
Planar curvature	0.815	1.226
TWI	0.958	1.044
NDVI	0.916	1.092
Precipitation	0.644	1.553
Distance from fault	0.756	1.271
Lithology	0.691	1.446

**Table 2 sensors-19-02685-t002:** Contingency matrix applied to the LSM, considering the average value of V_slope_ in each cell (the susceptibility degree from 1 to 5 represents very low, low, moderate, high, and very high, respectively).

	V_slope_ (mm/yr)					
Susceptibility degree		0–15	15–30	30–50	50–100	>100
	1	0	+1	+2	+3	+4
	2	0	0	+1	+2	+3
	3	0	0	0	+1	+2
	4	0	0	0	0	+1
	5	0	0	0	0	0

**Table 3 sensors-19-02685-t003:** Results of logistic regression analysis.

	B	S.E.	Wald	Sig.
Elevation	−0.928	0.040	551.073	0.000
Slope	1.022	0.032	991.807	0.000
Aspect	0.080	0.023	12.127	0.000
Profile curvature	0.298	0.032	89.027	0.000
Planar curvature	−0.125	0.025	24.136	0.000
TWI	0.069	0.034	4.151	0.042
NDVI	−0.308	0.049	38.809	0.000
Precipitation	0.496	0.034	215.025	0.000
Distance from fault	−0.566	0.036	246.496	0.000
Lithology	−0.021	0.017	1.520	0.218
Constant	−1.126	0.272	17.182	0.000

**Table 4 sensors-19-02685-t004:** Comparison of the original landslide susceptibility assessment degrees with the revised degrees after correction.

Landslide Susceptibility Degree	Original LSM		New LSM		Susceptibility Degree Increase	
Class	No. Cells	%	No. Cells	%	Class	No. Cells
1	481,668	57.11	480,041	56.91	0	840,628
2	166,190	19.70	167,670	19.88	+1	1387
3	91,057	10.80	90,059	10.68	+2	1084
4	56,650	6.71	57,130	6.77	+3	305
5	47,880	5.68	48,536	5.76	+4	32

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
