# Peer review of "Landslide Susceptibility Mapping of Karakorum Highway Combined with the Application of SBAS-InSAR Technology"

_sensors, 2019, doi:10.3390/s19122685_

Reviewer 1 Report

The present paper deals with using two different models, the logistic regression and the Random Forest for monitoring landslides. After the comparison of those approaches a classification map produced revealing high risk of landslides in two particular areas. In order to optimize the LSM result, the authors estimate the surface deformation using the SBAS InSAR. 

In my opinion, this work is very interesting as it shows the potential how landslides can be monitored, mitigated and eventually prevented. However, I think that the authors should work further on this paper. In my opinion, the weakest part of this work is the Interferometric analysis. The authors wrote that they used sentinel-1A images. Why not 1B as well with temporal separation of 6 days? I would recommend to show the interferograms that formed with their baselines and temporal separation. Another thing that it is missing in the InSAR analysis is the validation. The authors mentioned that the SBAS result is reliable, how they came to this statement? The authors also mentioned that did some field work, does this mean that you have GPS measurements available? If yes, it would be important to validate the SBAS result with those. For example, deformation histories. 

In Figure 9, there is a lack of information in the SBAS result. Is this area moving very fast? Please explain.

Additionally, I would expect a correlation between rainfall and the deformation rates. For example, did the authors notice any seasonal pattern which can be linked with the rainfall?

Some minor corrections:

L37: results instead of products 

L46: level instead of degrees

L49: caused instead of had 

L50: resulting instead of caused 

L66: for improving instead to improving 

L67: limitations of time...what do the authors mean with that? Avoid the long temporal separation?

L70: why only non linear? They method can monitor both linear and non linear deformation. What deformation model did the authors use?

L139: how the authors end up with the 100m grid. Please explain.

L142: 3.2 not 3.1 

L196: early warning and evolution of landslides?

When referring to SBAS method, the authors should add more references related to the technique itself, e.g. Berardino (2002).

Author Response

Point 1: The weakest part of this work is the Interferometric analysis. The authors wrote that they used sentinel-1A images. Why not 1B as well with temporal separation of 6 days? I would recommend to show the interferograms that formed with their baselines and temporal separation.

Response 1: We agree that would potentially be more useful to combine Sentinel 1A and 1B images with a temporal separation of 6 days. However, due to the later launching time (May 2016) and lack of Sentinel 1B in our study area and period, we only applied Sentinel 1A data for displacement generation. In addition, acquisitions at a 12-day interval are sufficient for slow deformation monitoring along the KKH. Because the aim of this research is to assess the landslide susceptibility of the KKH, refined by InSAR results, it would be a major task to process both Sentinel 1A and 1B data which would limit the efficiency of susceptibility assessment.

See lines 287-288.

As the reviewer suggested, we have added the connection graph of interferograms in the revised manuscript. See Figure 3.

Point 2: It is missing in the InSAR analysis is the validation. The authors mentioned that the SBAS result is reliable, how they came to this statement? The authors also mentioned that did some field work, does this mean that you have GPS measurements available? If yes, it would be important to validate the SBAS result with those. For example, deformation histories.

Response 2: Many thanks for your comments. Due to the complexity of the geological environment and the lack of monitoring data from long-term ground level measurements, it is difficult to validate the InSAR results with ground monitoring data. However, some previous studies have shown that the reliability of InSAR results could demonstrated by field survey (Wasowski and Bovenga, 2014; Zhang et al, 2016). Also, a detailed field survey shows that the activity of slopes along the KKH is consistent with displacement rates generated by SBAS-InSAR. Moreover, the InSAR results are reliable for the optimization of LSM (Figure 7b). See lines 438-442.

Wasowski J , Bovenga F . Investigating landslides and unstable slopes with satellite multi temporal interferometry; current issues and future perspectives[J]. Engineering Geology, 2014, 174(8):103-138.

Zhang Y , Meng X , Chen G , et al. Detection of geohazards in the Bailong River Basin using synthetic aperture radar interferometry[J]. Landslides, 2016, 13(5):1273-1284.

Point 3: In Figure 9, there is a lack of information in the SBAS result. Is this area moving very fast? Please explain.

Response 3: Thanks for your comments. The reason is that the deformation caused by mining is too rapid and exceeds the ability of the InSAR technique and Sentinel 1A datasets to resolve it. However, as stated by another reviewer, this paper is mainly focused on the evaluation of landslide susceptibility, and land subsidence caused by human activities should not be considered. Therefore, Figure 9 was deleted.

Point 4: Additionally, I would expect a correlation between rainfall and the deformation rates. For example, did the authors notice any seasonal pattern which can be linked with the rainfall?

Response 4: There is a strong correlation between rainfall and time series results of SBAS deformation. However, the main purpose of the paper is to optimize LSM using SBAS results. If rainfall analysis is added, the length of the paper would be greatly increased, which would reduce the emphasis on the main research topic. Therefore, the correlation between rainfall and deformation rates was not analyzed.

Some minor corrections:

L37: results instead of products

According to the reviewer’s suggestion, we have changed the form of expression.

See line 43.

L46: level instead of degrees

According to the reviewer’s suggestion, we have changed the form of expression.

See line 54.

L49: caused instead of had

According to the reviewer’s suggestion, we have changed the form of expression.

See line 58.

L50: resulting instead of caused

According to the reviewer’s suggestion, we have changed the form of expression.

See line 59.

L66: for improving instead to improving

According to the reviewer’s suggestion, we have changed the form of expression.

See line 99.

L67: limitations of time...what do the authors mean with that? Avoid the long temporal separation?

Yes, SBAS-InSAR overcomes the limitations of time incoherence and avoids long temporal separation.  See lines 104-105.

L70: why only non linear? They method can monitor both linear and non linear deformation. What deformation model did the authors use?

The SBAS-InSAR method is suitable for monitoring both linear and non-linear deformation, but the advantage of SBAS is that it has a better monitoring ability for nonlinear deformation like landslides. See lines 107-108.

In this paper, a cubic inversion model was used obtain the deformation results, because this model is more effective for monitoring landslides. See line 298.

L139: how the authors end up with the 100m grid. Please explain.

In this paper, the choice of the 100-m grid is based on both the resolution of the landslide predisposition factors and the mapping unit. See lines 205-211.

L142: 3.2 not 3.1

We have modified this. See line 216

L196: early warning and evolution of landslides?

I am sorry but we can’t find this sentence.

When referring to SBAS method, the authors should add more references related to the technique itself, e.g. Berardino (2002).

Thank you for providing some references for us. We have added references related to the SBAS method itself.  See line107.

Reviewer 2 Report

Dear authors,

The present manuscript comprise the assessment of landslide susceptibility along the Karakorum Highway using two different models (logistic regression and random forest) and combine the SBAS-InSAR terrain deformation data along the slope direction with the Random Forest landslide susceptibility model to improve the initial landslide susceptibility map. The manuscript is well structured and methodological supported; nevertheless the analysis of the results, particularly those related to SBAS-InSAR and discussion should be substantially improved. The manuscript presents some important fragilities in my opinion or aspects that must be clarified and consequently should not be published in its present form. My recommendation is that the manuscript needs major revisions. The most relevant drawbacks/fragilities and doubts along the manuscript to the reviewer are summarized in the following general comments and in the particular remarks section.

Although I recognize the regional importance of the work, authors should clarify better the novelty of the work from the methodological point of view, when comparing for example with other similar works in literature. Which gaps the work intends to fulfill.

A detailed description of the landslide inventory is necessary. A description of landslide types, geometry, triggering conditions, slope context where they occur (e.g., road cuts/fills, landslides due to bank erosion) or landslide feature type (e.g., polygon, point, landslide area, landslide depletion area) are critical issues regarding the LSM that must be addressed in the manuscript.

Authors should include in the manuscript a new figure with the dataset of variables used as landslide predisposing factors in the landslide susceptibility models.

The Results and Discussion sections need to be reformulated according my particular comments to authors. Authors must address better the relation between the Persistent Scatterers (PS) terrain deformation measurements and landslide occurrence. In my modest opinion the refining approach used by to improve the landslide susceptibility map is only valuable if PS deformation rates are related, specifically, with landslide occurrence/deformation. ~

The conclusions section is too long and should be substantially reduced. In my opinion authors should reformulate the conclusions text in order to be away from a simple summary of the main results.

----------------------------------------

Particular remarks

Section. Abstract

Line 24: “…high risk of landsliding…” although I understand the meaning of authors statement is preferable to mention susceptibility instead of risk. In fact no risk assessment is made in the present work.

Lines 30-33: This last sentence of the abstract should be reformulated according suggestions made along the specific sections of the manuscript.

------------------------------

Section 1. Introduction

Lines 54-62: The paragraph is too generic, and in my opinion not focused as it could be in the state of the art related to LSM. Please improve description.

Lines 70-72: The use of PSinSAR techniques for landslide susceptibility assessment or improvement of landslide susceptibility maps should be better addressed in this part of the introduction. Please verify specific literature, and adjust the text accordingly, for example:

Piacentini, D., Devoto, S., Mantovani, M. et al. Nat Hazards (2015) 78: 681. https://doi.org/10.1007/s11069-015-1740-8

Oliveira, S.C., Zêzere, J.L., Catalão, J. et al. Landslides (2015) 12: 703. https://doi.org/10.1007/s10346-014-0522-9

Rott, H., Nagler, T. Advances in Space Research (2006) 37 (4): 710-719 https://doi.org/10.1016/j.asr.2005.06.059

Lines 77-80: Authors say: ”The approach was found to be very successful for evaluating the landslide susceptibility level in the study area. However, there was still potential for the misclassification of the results of the LSM: for example, if a landslide occurs in a certain area, the susceptibility map may still indicate that the area is stable“. Please rewrite. The first sentence is related with results and should not be used in introduction. In addition, the approach used by authors do not classify the terrain units into stable or unstable, only defines the susceptibility potential of each terrain unit of the study area in a hierarchic way, from the most susceptible to the less susceptible.

------------------------------

Section 3 (Data and methods)

3.1. Data and variables

Line 108: Please consider to replace “build” by “training”. Adjust text if necessary. If related, please consider using training instead of build along the manuscript.

Lines 135-141: Please consider to rewrite the paragraph in a simple manner. In my opinion is not clear. For me is not clear why choose a 100m grid cell and why include in the end of the paragraphs the statistical methods. The size of the grid cell should be reflected in all variables and methods, form the dataset of variables assumed as predisposing factors to the landslide inventory, independently of the method. In the case of the landslide inventory, each point should be the centroid of each 100m size pixel. Is that? Please turn clear.

------------------------------

Section 3.1 (Landslide susceptibility models)

Line 142: Please verify the number of the section “Landslide susceptibility models”. It should be 3.2

------------------------------

Section 3.2.1 (Logistic regression model)

Line 149: Please verify “In geological hazard…”. It should be better to say susceptibility. The assessment of the hazard component should include the recurrence of the triggering factor.

------------------------------

Section 3.2.3. Verification of model accuracy

Lines 178-184: Please see my comment to manuscript lines 109-111 in section 3.1. Data and variables. This topic should incorporate possible bias discussion in the discussion section.

------------------------------

Section 3.4. Refinement

Lines 237-239: “This integration reduces the probability of misclassification, and it results in areas prone to landslides which have a low susceptibility assessment degree being assigned a high susceptibility degree”. I agree with authors but this point should be effectively discussed in the discussion section. In fact, some of the causes for the misclassification among others are related with the quality of data associated to each landslide predisposing factor and the other one is the quality of the landslide inventory. Regarding the last one, for this study area exists only 44 landslides mapped and no description about those landslide inventory is made by authors. It is a complete landslide inventory? Maybe not, otherwise all deformation capture by SBAS-InSAR should be inside the landslide limits. This should be take into account in the discussion.

------------------------------

Section 4. Results

Section 4.1. Logistic regression results

Lines 256-257: Please verify last sentence, it’s not correct in my opinion. As far as could understand, authors are using a citation to the work of (Dai and Lee 2002) made by [33]. The original text in reference [33] is “when the thickness of soil layer is sufficient, the higher degrees of slope is, the more unstable the slope is“. For me “more unstable” is more appropriate than “greater instability”.

------------------------------

Section 4.2. Random Forest results

Lines 296-312: Please verify paragraph. The description made by authors about the different types of landslides that occur in each study area reveal a potential major drawback of this work, and that is related with the landslide inventory map. Different types of landslides such as debris flows or landslides imply different modelling approaches. For the first ones the implication is to model rupture and propagation and for the second ones only rupture (if regarding the geometry of the rupture surface doesn’t occur different types of landslide occurring in sequence, e.g., slide + flow)

-----------------------------

Section 4.3. SBAS InSAR results

Line 319: with “deformation points” authors refer to persistent Scatterers? If so adjust change along the manuscript.

------------------------------

Section 4.4. Refining the results

Line 313: Please verify the number of the section “. It should be 4.4?

Lines 341-342: “There are three areas with obvious landslides with increased landslide susceptibility”. This section should be reformulated. In fact, and in my modest opinion, authors should be more critic regarding deformation data obtained by the SBAS PSInSAR technique. As pointed out by authors and giving the more critic example, that reports to deformation data associated to locality 3, the main process is related to land subsidense related to mining activity, therefore, it cannot be understood as a landslide and should be excluded from this refinement process.

------------------------------

Section 4.5. Results of specific cases

Line 313: Please verify the number of the section “. It should be 4.5?

Lines 359-360: Authors say: “Locality 1, on the west side of Blumkou Reservoir, is a hilly area composed mainly of loose sand deposits. It is substantially affected by the action of wind and rain, and the sandy material is accumulating rapidly in the reservoir.” Please be more specific about the relation of the described erosion processes with landslide occurrence. If the erosion is not related to landslides the deformation values should not be used for increase the susceptibility degree in the new landslide map.

Line 378: I completely agree with authors but I believe that the most important point is still missing from the discussion. What is the meaning of the acceleration of the terrain deformation regarding landslide susceptibility? What it represents from the statistical relationships between predisposing factors and landslide points? It means an incomplete landslide inventory map?

Lines 385-388: Why authors use deformation data, which is related to ground subsidence caused by mining activity? This point is quite surprising to me! What is the relation with landslides? My understanding is that subsidence is in fact a terrain movement but not a landslide, therefore the terrain deformation data should not be used to refining the landslide susceptibility map.

------------------------------

Section 5. Discussion

Line 410-413: Authors say “Notably, the new LSM, combined with the SBAS results, reduced the misclassification in which terrain affected by ground deformation was classified as stable. The deformation includes not only potential landslides, but also other forms of slope instability and land subsidence”. Please reformulate this text section. The classification approach used by authors does not classify the terrain units as stable but only as very low and low susceptibility. Another important point in my opinion is that authors should only use Persistent scatterers for which terrain deformation values are related to landslide occurrence.

------------------------------

Section 6. Conclusions

Line 433: Authors say: ”A 10 km buffer zone along the KKH was selected…”. As far as it was possible to me to understand this information is presented for the first time in conclusions section. This information should be referred in the section “study area”. Moreover, what was the criteria do select the 10 km buffer? Is related to landslide distribution? What could be the bias introduced in the susceptibility models by this option? That should be discussed by authors in the discussion section.

------------------------------

Comments on Figures and Tables

Figure 1

Figure 1 should be substantially improved. The landslide inventory is not visible. Please consider to adopt other feature for representing landslides rather than landslide area. Authors should also consider to add additional zoom windows to areas with landslides. Moreover, red dots representing the location of cities should not have the same color has landslides. Change the color of these dots also for all the other maps. Blue, red and yellow lines over the map are not included in the legend of figure what is their meaning? Authors should consider to use a higher font size for text along figure.

Names as Gaizi Valley and Tashkurgan Valley as other places mentioned for which authors report specific situations along the text must be identified in Figure 1.

Figure 2

ROC curve figure is too small. Please consider to enlarge it. Again consider to use a higher font size for text along figure.

Figure 3

Please consider to enlarge figure.

Figure 5

Enlarge figure or consider to use a higher font size for text along figure.

Figure 6

Figure 6 should be completely reformulated based on comments along text. Mostly authors consider the area of mining activity as suitable for refining the landslide susceptibility map and should not. In fact, subsidence is considering a terrain movement but not a landslide occurring along a slope. Therefore, this test site should not be considering. Removing this test site means in my modest opinion to reduce substantially the importance of the approach for this study area.

Please increase the size of numbers 1-3 that represent each locality in figure 6b.

Figures 7 and 8

Please increase the letter cases in figure. Legends text is not easy to read.

Figure 9

My suggestion is that Figure 9 should be eliminated. It do not relates to landslide processes.

Table 2

The velocity classes of 50-100 and > 100 are not underlined in table. Please report along the manuscript in the Refinement section the criteria used to classify the velocity of deformation in these 5 classes. Why these break limits?

Table 3

Authors should consider to separate the “constant” values from the order of the variables list.

Table 4

The column titles New LSM (No. cells, %) and Increase (class, No. cells) are not underlined in table. The title of column “increase” should be defined by authors in a more specific way. Moreover, the differences between the number of cells in the original LSM and the new LSM are not these represented in column “Increase”. Please verify and if necessary adjust the text along the manuscript, accordingly. In addition, I do not understand the number of cells of 836823 in the column “increase” for the class 1

Author Response

Point 1: Although I recognize the regional importance of the work, authors should clarify better the novelty of the work from the methodological point of view, when comparing for example with other similar works in literature. Which gaps the work intends to fulfill.

Response 1: Previous research focused on the investigation and evaluation of a single disaster. This study applies the InSAR method to optimize landslide susceptibility mapping, which not only compensates for the unreliability of landslide susceptibility mapping caused by an insufficient number of historical landslides, but it also fills the gap of landslide susceptibility mapping on the regional scale in the KKH. See lines 64-75.

Point 2: A detailed description of the landslide inventory is necessary. A description of landslide types, geometry, triggering conditions, slope context where they occur (e.g., road cuts/fills, landslides due to bank erosion) or landslide feature type (e.g., polygon, point, landslide area, landslide depletion area) are critical issues regarding the LSM that must be addressed in the manuscript.

Response 2: The landslide inventory in this study is based on the interpretation of high-resolution images, which includes landslide features (polygons, points, landslide areas), and are sufficient for landslide susceptibility assessment in this case study. Other characteristics, such as types and triggers, are not necessary for landslide assessment. Otherwise, it is difficult to make a very detailed description of historical landslides due to few published studies. See lines152-159, Figure 1.

Point 3: Authors should include in the manuscript a new figure with the dataset of variables used as landslide predisposing factors in the landslide susceptibility models.

Response 3: We have added a new figure with the dataset of variables used as landslide predisposition factors in the landslide susceptibility models. See Figure 2.

Point 4: The Results and Discussion sections need to be reformulated according my particular comments to authors. Authors must address better the relation between the Persistent Scatterers (PS) terrain deformation measurements and landslide occurrence. In my modest opinion the refining approach used by to improve the landslide susceptibility map is only valuable if PS deformation rates are related, specifically, with landslide occurrence/deformation. 

Response 4: Yes, the optimization of landslide susceptibility mapping is only effective when the SBAS deformation indicates landslide occurrence. In fact, the slope deformation recognized by the InSAR method is generally a precursor to landslide occurrence. When the slope deformation rate has an accelerated process in time series analysis, it generally indicates the occurrence of landslide (Herrera G,2013; Zhang, Yi, 2018). Through field investigation and verification of InSAR results, we found that the landslide state of activity has been determined through Vslope; that is, the greater the value of Vslope, the more unstable is the slope and the greater the likelihood of a landslide occurring. It is demonstrated that InSAR results can greatly improve the monitoring capacity of very slow landslides. Thus, deformation generated from InSAR can provide important evidence for the early identification and susceptibility evaluation of landslides (Oliveira,2014; Piacentini,2015).

Herrera G , F. Gutiérrez, J.C. García-Davalillo, et al. Multi-sensor advanced DInSAR monitoring of very slow landslides: The Tena Valley case study (Central Spanish Pyrenees)[J]. Remote Sensing of Environment, 2013, 128(none):31---43.

Zhang, Yi, Xingmin Meng, Colm Jordan, Alessandro Novellino, Tom Dijkstra, and Guan Chen. "Investigating Slow-Moving Landslides in the Zhouqu Region of China Using Insar Time Series." Landslides 15, no. 7 (2018): 1299-315.

Oliveira, S. C., J. L. Zêzere, J. Catalão, and G. Nico. "The Contribution of Psinsar Interferometry to Landslide Hazard in Weak Rock-Dominated Areas." Landslides 12, no. 4 (2014): 703-19.

Piacentini, Daniela, Stefano Devoto, Matteo Mantovani, Alessandro Pasuto, Mariacristina Prampolini, and Mauro Soldati. "Landslide Susceptibility Modeling Assisted by Persistent Scatterers Interferometry (Psi): An Example from the Northwestern Coast of Malta." Natural Hazards 78, no. 1 (2015): 681-97.

See lines 562-572.

Point 5: The conclusions section is too long and should be substantially reduced. In my opinion authors should reformulate the conclusions text in order to be away from a simple summary of the main results.

Response 5: We have rewritten the conclusions section according to the Reviewer’s comments. See lines 614-665.

Particular remarks

Section. Abstract

Line 24: “…high risk of landsliding…” although I understand the meaning of authors statement is preferable to mention susceptibility instead of risk. In fact no risk assessment is made in the present work.

According to reviewer’s suggestion, we have changed the form of expression.

See line 26.

Lines 30-33: This last sentence of the abstract should be reformulated according suggestions made along the specific sections of the manuscript.

According to the reviewer’s suggestion, we have rewritten the last sentence of the abstract. See lines 35-38.

------------------------------

Section 1. Introduction

Lines 54-62: The paragraph is too generic, and in my opinion not focused as it could be in the state of the art related to LSM. Please improve description.

According to reviewer’s suggestion, we have rewritten the paragraph. See lines 76-93.

Lines 70-72: The use of PSinSAR techniques for landslide susceptibility assessment or improvement of landslide susceptibility maps should be better addressed in this part of the introduction. Please verify specific literature, and adjust the text accordingly, for example:

Piacentini, D., Devoto, S., Mantovani, M. et al. Nat Hazards (2015) 78: 681. https://doi.org/10.1007/s11069-015-1740-8

Oliveira, S.C., Zêzere, J.L., Catalão, J. et al. Landslides (2015) 12: 703. https://doi.org/10.1007/s10346-014-0522-9

Rott, H., Nagler, T. Advances in Space Research (2006) 37 (4): 710-719 https://doi.org/10.1016/j.asr.2005.06.059

In this paper, we used SBAS-InSAR to obtain the slope deformation rate and optimize the LSM, because the SBAS method is more suitable for extracting slope deformation in mountainous area (参考文献). According to the reviewer’s suggestion, we have added some specific references to demonstrate the value of InSAR for landslide susceptibility assessment and LSM improvement. See lines 94-103.

Lines 77-80: Authors say: ”The approach was found to be very successful for evaluating the landslide susceptibility level in the study area. However, there was still potential for the misclassification of the results of the LSM: for example, if a landslide occurs in a certain area, the susceptibility map may still indicate that the area is stable“. Please rewrite. The first sentence is related with results and should not be used in introduction. In addition, the approach used by authors do not classify the terrain units into stable or unstable, only defines the susceptibility potential of each terrain unit of the study area in a hierarchic way, from the most susceptible to the less susceptible.

According to the reviewer’s suggestion, we have rewritten the sentence. See lines 116-120

------------------------------

Section 3 (Data and methods)

3.1. Data and variables

Line 108: Please consider to replace “build” by “training”. Adjust text if necessary. If related, please consider using training instead of build along the manuscript.

Yes, “training” is better than “build”, we have changed in the manuscript. See line161, 165.

Lines 135-141: Please consider to rewrite the paragraph in a simple manner. In my opinion is not clear. For me is not clear why choose a 100m grid cell and why include in the end of the paragraphs the statistical methods. The size of the grid cell should be reflected in all variables and methods, form the dataset of variables assumed as predisposing factors to the landslide inventory, independently of the method. In the case of the landslide inventory, each point should be the centroid of each 100m size pixel. Is that? Please turn clear.

Yes, we have rewritten the paragraph in a simpler manner. The size of the grid cell is also the mapping unit, when the landslide predisposition factors were determined; the next step is to define the mapping unit for both these factors and the LSM. Therefore, we include this section at the end of the paragraphs.

It is correct that in the case of the landslide inventory, each point should be the centroid of each 100-m pixel. See lines 205-212.

------------------------------

Section 3.1 (Landslide susceptibility models)

Line 142: Please verify the number of the section “Landslide susceptibility models”. It should be 3.2

We have modified this. See line 216.

------------------------------

Section 3.2.1 (Logistic regression model)

Line 149: Please verify “In geological hazard…”. It should be better to say susceptibility. The assessment of the hazard component should include the recurrence of the triggering factor.

We have changed the form of expression. See line 225

------------------------------

Section 3.2.3. Verification of model accuracy

Lines 178-184: Please see my comment to manuscript lines 109-111 in section 3.1. Data and variables. This topic should incorporate possible bias discussion in the discussion section.

I am sorry but we can’t find the comments to the paper. In this section, we have added discussion of the possible bias in the discussion section. See lines 550-561.

------------------------------

Section 3.4. Refinement

Lines 237-239: “This integration reduces the probability of misclassification, and it results in areas prone to landslides which have a low susceptibility assessment degree being assigned a high susceptibility degree”. I agree with authors but this point should be effectively discussed in the discussion section. In fact, some of the causes for the misclassification among others are related with the quality of data associated to each landslide predisposing factor and the other one is the quality of the landslide inventory. Regarding the last one, for this study area exists only 44 landslides mapped and no description about those landslide inventory is made by authors. It is a complete landslide inventory? Maybe not, otherwise all deformation capture by SBAS-InSAR should be inside the landslide limits. This should be take into account in the discussion.

Yes, we have added discussion about the possible reasons for misclassification and the necessity of optimization using InSAR. See lines 584-603.

------------------------------

Section 4. Results

ection 4.1. Logistic regression results

Lines 256-257: Please verify last sentence, it’s not correct in my opinion. As far as could understand, authors are using a citation to the work of (Dai and Lee 2002) made by [33]. The original text in reference [33] is “when the thickness of soil layer is sufficient, the higher degrees of slope is, the more unstable the slope is“. For me “more unstable” is more appropriate than “greater instability”.

Yes, we have changed the form of expression. See lines 356-357.

------------------------------

Section 4.2. Random Forest results

Lines 296-312: Please verify paragraph. The description made by authors about the different types of landslides that occur in each study area reveal a potential major drawback of this work, and that is related with the landslide inventory map. Different types of landslides such as debris flows or landslides imply different modelling approaches. For the first ones the implication is to model rupture and propagation and for the second ones only rupture (if regarding the geometry of the rupture surface doesn’t occur different types of landslide occurring in sequence, e.g., slide + flow)

We have changed some expressions in this paragraph. This study mainly focuses on the analysis of landslides, including rock falls, unstable slope, creep, etc. Without considering flow, additionally, the SBAS method can only obtain the deformation of a slide, since the flow deformation is too rapid and  InSAR cannot obtain effective monitoring values. See lines 408-410.

-----------------------------

Section 4.3. SBAS InSAR results

Line 319: with “deformation points” authors refer to persistent Scatterers? If so adjust change along the manuscript.

In this paper, we used the SBAS-InSAR method to obtain the deformation points, not by PS-InSAR. The deformation points are the coherent pixels in  hich Vslope is less than -15 mm/yr, and also the points which have an apparent deformation, not consistent with the PS. 

------------------------------

Section 4.4. Refining the results

Line 313: Please verify the number of the section “. It should be 4.4?

We have modified this. See line 457.

Lines 341-342: “There are three areas with obvious landslides with increased landslide susceptibility”. This section should be reformulated. In fact, and in my modest opinion, authors should be more critic regarding deformation data obtained by the SBAS PSInSAR technique. As pointed out by authors and giving the more critic example, that reports to deformation data associated to locality 3, the main process is related to land subsidense related to mining activity, therefore, it cannot be understood as a landslide and should be excluded from this refinement process.

In this paper, when conducting the optimization of LSM, the deformation results of SBAS delineated as subsidence were removed; then we used the new Vslope after removing the subsidence information and LSM obtained by the Random Forest model to optimize. The analysis of locality 3 was deleted. See lines 447-451.

------------------------------

Section 4.5. Results of specific cases

Line 313: Please verify the number of the section “. It should be 4.5?

We have modified this. See line 485.

Lines 359-360: Authors say: “Locality 1, on the west side of Blumkou Reservoir, is a hilly area composed mainly of loose sand deposits. It is substantially affected by the action of wind and rain, and the sandy material is accumulating rapidly in the reservoir.” Please be more specific about the relation of the described erosion processes with landslide occurrence. If the erosion is not related to landslides the deformation values should not be used for increase the susceptibility degree in the new landslide map.

We have added a detailed description of the relationship of the described erosion processes with landslide occurrence. The erosion is related to landslides, and the deformation values should be used to  increase the susceptibility degree in the new LSM. See lines 489-496.

Line 378: I completely agree with authors but I believe that the most important point is still missing from the discussion. What is the meaning of the acceleration of the terrain deformation regarding landslide susceptibility? What it represents from the statistical relationships between predisposing factors and landslide points? It means an incomplete landslide inventory map?

 We have changed the form of expression. As we lack a detailed landslide inventory map, some landslide-prone areas are misclassified into low susceptibility areas, which makes it necessary to optimize the LSM by using SBAS results. The apparent SBAS deformation rate has a significance in landslide susceptibility. See lines 516-522, 562-572, 584-598.

Lines 385-388: Why authors use deformation data, which is related to ground subsidence caused by mining activity? This point is quite surprising to me! What is the relation with landslides? My understanding is that subsidence is in fact a terrain movement but not a landslide, therefore the terrain deformation data should not be used to refining the landslide susceptibility map.

Yes, the subsidence analysis shouldn’t be considered in the LSM. So we have deleted the related analysis and use the Vslope after removing the subsidence deformation to optimize the LSM. 

------------------------------

Section 5. Discussion

Line 410-413: Authors say “Notably, the new LSM, combined with the SBAS results, reduced the misclassification in which terrain affected by ground deformation was classified as stable. The deformation includes not only potential landslides, but also other forms of slope instability and land subsidence”. Please reformulate this text section. The classification approach used by authors does not classify the terrain units as stable but only as very low and low susceptibility. Another important point in my opinion is that authors should only use Persistent scatterers for which terrain deformation values are related to landslide occurrence.

We have rewritten this section. See lines 594-596.

Yes, we should only consider the deformation rates related to landslide occurrence, not including the subsidence information. See lines 447-451.

------------------------------

Section 6. Conclusions

Line 433: Authors say: ”A 10 km buffer zone along the KKH was selected…”. As far as it was possible to me to understand this information is presented for the first time in conclusions section. This information should be referred in the section “study area”. Moreover, what was the criteria do select the 10 km buffer? Is related to landslide distribution? What could be the bias introduced in the susceptibility models by this option? That should be discussed by authors in the discussion section.

Yes, we have added a description of the study area. See lines 125.

We have added a discussion about the choice of a 10-km buffer zone. See lines 541-549.

------------------------------

Comments on Figures and Tables

Figure 1

Figure 1 should be substantially improved. The landslide inventory is not visible. Please consider to adopt other feature for representing landslides rather than landslide area. Authors should also consider to add additional zoom windows to areas with landslides. Moreover, red dots representing the location of cities should not have the same color has landslides. Change the color of these dots also for all the other maps. Blue, red and yellow lines over the map are not included in the legend of figure what is their meaning? Authors should consider to use a higher font size for text along figure.

Names as Gaizi Valley and Tashkurgan Valley as other places mentioned for which authors report specific situations along the text must be identified in Figure 1.

We have modified the figure as suggested and added the main places mentioned in this paper.

See Figure 1

Figure 2

 ROC curve figure is too small. Please consider to enlarge it. Again consider to use a higher font size for text along figure.

We have modified the figure as the reviewer suggested. See Figure 4. 

Figure 3

Please consider to enlarge figure.

We have modified the figure as the reviewer suggested. See Figure 5.

Figure 5

Enlarge figure or consider to use a higher font size for text along figure.

We have modified the figure as the reviewer suggested. See Figure 7.

Figure 6

Figure 6 should be completely reformulated based on comments along text. Mostly authors consider the area of mining activity as suitable for refining the landslide susceptibility map and should not. In fact, subsidence is considering a terrain movement but not a landslide occurring along a slope. Therefore, this test site should not be considering. Removing this test site means in my modest opinion to reduce substantially the importance of the approach for this study area.

Please increase the size of numbers 1-3 that represent each locality in figure 6b.

We have modified the figure as the reviewer suggested. The Vslope results related to subsidence are removed in the LSM optimization, and the specific analysis of subsidence is also removed. See Figure 8.

Figures 7 and 8

Please increase the letter cases in figure. Legends text is not easy to read.

We have modified the figure as the reviewer suggested. See Figure 9, 10.

Figure 9

My suggestion is that Figure 9 should be eliminated. It do not relates to landslide processes.

We have deleted the figure as the reviewer suggested

Table 2

The velocity classes of 50-100 and > 100 are not underlined in table. Please report along the manuscript in the Refinement section the criteria used to classify the velocity of deformation in these 5 classes. Why these break limits?

We have modified the table as the reviewer suggested. See Table 2.

The criteria used to classify the deformation velocity is calculated based on the standard deviation of all points’ Vslope. The KKH is geologically very active, and the slope deformation is no longer slow or extremely slow, and the existing standards of active limits cannot satisfy the division of deformation rate. Also, the standards based on the velocity of LOS direction are not suitable for Vslope, and therefore we just used the statistical result to classify the deformation velocity. See lines 327-329. 

Table 3

Authors should consider to separate the “constant” values from the order of the variables list.

We have modified the table as the reviewer suggested. See Table 3.

Table 4

The column titles New LSM (No. cells, %) and Increase (class, No. cells) are not underlined in table. The title of column “increase” should be defined by authors in a more specific way. Moreover, the differences between the number of cells in the original LSM and the new LSM are not these represented in column “Increase”. Please verify and if necessary adjust the text along the manuscript, accordingly. In addition, I do not understand the number of cells of 836823 in the column “increase” for the class 1

We have modified the table as the reviewer suggested. See Table 4

The new table is based on the new integrated LSM obtained by the Random Forest model and Vslope after removing the subsidence deformation points. 840,628 is the number of cells in which the susceptibility degree has not changed: that is the number of cells of increase 0, and the number of cells with a susceptibility degree of 1 is 480, 041 after optimization

Round  2

Reviewer 1 Report

Dear authors,

Thank you for your effort and replying to my comments. In my opinion, the manuscript is ready now for publication.